# Genome-Wide Super-Enhancer-Based Analysis: Identification of Prognostic Genes in Oral Squamous Cell Carcinoma

**DOI:** 10.3390/ijms23169154

**Published:** 2022-08-15

**Authors:** Tomoaki Saito, Shunichi Asai, Nozomi Tanaka, Nijiro Nohata, Chikashi Minemura, Ayaka Koma, Naoko Kikkawa, Atsushi Kasamatsu, Toyoyuki Hanazawa, Katsuhiro Uzawa, Naohiko Seki

**Affiliations:** 1Department of Oral Science, Graduate School of Medicine, Chiba University, Chiba 260-8670, Japan; 2Department of Functional Genomics, Chiba University Graduate School of Medicine, Chiba 260-8670, Japan; 3Department of Otorhinolaryngology/Head and Neck Surgery, Chiba University Graduate School of Medicine, Chiba 260-8670, Japan; 4MSD K.K., Tokyo 102-8667, Japan

**Keywords:** oral squamous cell carcinoma, super-enhancers, chromatin immunoprecipitation sequencing, H3K27ac, *C9orf89*, *CENPA*, *PISD*, *TRAF2*, TCGA-OSCC

## Abstract

Advanced-stage oral squamous cell carcinoma (OSCC) patients are treated with combination therapies, such as surgery, radiation, chemotherapy, and immunotherapy. However, OSCC cells acquire resistance to these treatments, resulting in local recurrence and distant metastasis. The identification of genes involved in drug resistance is essential for improving the treatment of this disease. In this study, we applied chromatin immunoprecipitation sequencing (ChIP-Seq) to profile active enhancers. For that purpose, we used OSCC cell lines that had been exposed to cetuximab for a prolonged period. In total, 64 chromosomal loci were identified as active super-enhancers (SE) according to active enhancer marker histone H3 lysine 27 acetylation (H3K27ac) ChIP-Seq. In addition, a total of 131 genes were located in SE regions, and 34 genes were upregulated in OSCC tissues by TCGA-OSCC analysis. Moreover, high expression of four genes (*C9orf89*; *p* = 0.035, *CENPA*; *p* = 0.020, *PISD*; *p* = 0.0051, and *TRAF2*; *p* = 0.0075) closely predicted a poorer prognosis for OSCC patients according to log-rank tests. Increased expression of the four genes (mRNA Z-score ≥ 0) frequently co-occurred in TCGA-OSCC analyses. The high and low expression groups of the four genes showed significant differences in prognosis, suggesting that there are clear differences in the pathways based on the underlying gene expression profiles. These data indicate that potential stratified therapeutic strategies could be used to overcome resistance to drugs (including cetuximab) and further improve responses in drug-sensitive patients.

## 1. Introduction

Oral cancer is a malignant neoplasm that begins in various parts of the oral cavity, e.g., the tongue, buccal mucosa, and the floor of the mouth. Most of them are oral squamous cell carcinomas (OSCC) [1]. According to Global Cancer Statistics 2018, there are approximately 350,000 new cases of OSCC and 180,000 deaths from the disease per year worldwide [2]. Metastatic and advanced stage cases of oral cancer have a poor prognosis, i.e., the 5-year survival rate is less than 50% [3]. Surgical resection is the first line of treatment of patients with OSCC. However, combinations of chemo-radiation therapy, molecularly targeted therapy and immunotherapy are selected for unresectable or advanced-stage cases [3,4]. First-line treatment regimens for recurrent or metastatic HNSCC include the immune checkpoint inhibitor pembrolizumab [5] and the EXTREME regimen, which includes cetuximab [6]. Cetuximab is an important treatment option as a second-line (and later) option for recurrent or metastatic HNSCC.

In general, cancer cells respond well to initial treatments, but develop resistance during ongoing treatment. There are few effective treatments for cancer cells that have acquired resistance to anticancer drugs (cisplatin; CDDP, 5-fluorouracil; 5-FU, and paclitaxel) or molecular targeted therapy (cetuximab) [7,8]. Identifying molecular events in cancer cells that lead to treatment resistance is an important goal in cancer research and represents significant challenges.

Various analytical approaches (gene expression profiles, noncoding RNA profiles, and chromosomal alteration) have been explored to elucidate the molecular mechanisms of drug resistance [8,9]. Previous studies have revealed that the expression of anticancer drug excretion genes, DNA repair genes, antiapoptotic genes, and epithelial mesenchymal transition (EMT)-related genes may be activated in anticancer drug-resistant cancer cells [10,11,12,13].

Recent studies have revealed that epigenetic factors (e.g., including DNA methylation, histone modifications, non-coding RNAs) are pivotal players in the malignant transformation of cancer cells [14,15,16]. Epigenetic changes in gene expression are reversible and do not alter the DNA sequence but may change the way the DNA sequence is read and expressed.

The binding of transcription factors to an enhancer is an important first step in gene expression [17]. Enhancers are short (50–1500 base pair) DNA regions to which transcription factors bind to increase the likelihood of transcription of a particular gene [18]. Super-enhancers are formed to strongly express pivotal genes that determine the fate of cells [19,20]. A super-enhancer (SE) is a region of the mammalian genome that contains multiple enhancers that collectively bind transcription factors to facilitate gene transcription [19,21]. Elevated enhancer activities are involved in the resistance of cancer cells following treatment with anticancer drugs [22,23,24,25].

Recent cancer research is clarifying the mechanism by which higher-order chromatin structures (modulated by DNA methylation and histone modification) are involved in human oncogenesis and drug resistance [26,27,28]. H3K27ac (acetylation of the lysine residue at N-terminal position 27 of the histone H3 protein) is associated with higher activation of transcription and is therefore defined as an active enhancer mark [29,30].

Chromatin immunoprecipitation sequencing (ChIP-Seq) is a well-established technology that combines chromatin immunoprecipitation (ChIP) with next-generation sequencing. By using this method, it is possible to analyze histone modification and the binding sites of transcriptional regulators (DNA-binding proteins) on the genome in a genome-wide manner [31,32]. ChIP-Seq analysis targeting H3K27ac reveals that a super enhancer in the genome is active [33,34].

In previous studies, we established cetuximab-resistant OSCC cell lines and identified several genes involved in cetuximab resistance [35]. In this study, we attempted to identify super-enhancers involved in drug resistance using OSCC cell lines that had been treated with cetuximab for prolonged periods. A total of 64 genomic loci of H3K27ac-related super-enhancers were identified by comparing the parental cell lines with the cetuximab-treated cell lines. A total of 131 genes were identified in the super-enhancers, and The Cancer Genome Atlas (TCGA) revealed that 34 genes were highly expressed in OSCC clinical tissues. Importantly, high expression of four genes (*C9orf89*; *p* = 0.035, *CENPA*; *p* = 0.020, *PISD*; *p* = 0.0051, and *TRAF2*; *p* = 0.0075) closely predicted a poorer prognosis of OSCC patients. Here, we provide information on super-enhancers involved in drug resistance. Identification of candidate genes should accelerate our understanding of the molecular mechanisms of drug resistance.

## 2. Results

### 2.1. Genome-Wide Screening of SE in OSCC Cell Lines following Long-Term Treatment with Cetuximab Using H3K27ac ChIP-Seq Analysis

To investigate the dynamic epigenetic state of OSCC after long-term exposure to anticancer drugs, we identified SEs with H3K27ac peaks in control cells (HSC-3 and SAS) and cells that had been subjected to long-term cetuximab exposure (Cmab-LTE). A total of 995 and 1043 SE peaks were detected in Cmab-LTE HSC-3 and Cmab-LTE SAS cells (Figure 1).

Next, we compared the control cells and the Cmab-LTE cells. A total of 152 and 481 SE gain peaks were identified in Cmab-LTE HSC-3 and Cmab-LTE SAS compared with the respective parental cell lines, respectively (Figure 1 and Figure 2). Of these, 68 gain peaks were common to the two cell lines. Analysis of a human genome database revealed that a total of 131 genes corresponded to those SE regions (Figure 2). The detailed information of the genes in the SE regions is shown in Table 1.

These genes might be prognostic markers in patients with OSCC. Thus, clinicopathological analysis of these genes in OSCC patients was performed using the TCGA-OSCC database (below). The 34 genes for which expression was upregulated by TCGA-OSCC tissues are shown in bold in the Table 1, and the details of the genes were listed in Appendix A.

### 2.2. Clinical Significance of C9orf89, CENPA, PISD, and TRAF2, in OSCC Patients Determined by TCGA-OSCC Analysis

TCGA-OSCC database analysis showed that a total of 34 genes were upregulated in OSCC tissues compared to normal tissues (Figure 2, Appendix A). Among these upregulated genes, expression of four genes (*C9orf89*, *CENPA*, *PISD*, and *TRAF2*) significantly predicted 5-year overall survival rates in OSCC patients (Figure 3).

Specific H3K27ac signals in Cmab-LTE cell lines (HSC-3 and SAS) are shown in Figure 4. Chromosomal regions located in *C9orf89*, *CENPA*, *PISD*, and *TRAF2* genes are shown in Table 1. Further analysis of these four genes was performed.

### 2.3. Expression Levels of C9orf89, CENPA, PISD, and TRAF2 in OSCC Cells after Long-Term Exposure to Cetuximab

We verified that gene expression levels were induced by cetuximab treatment. The expression levels of four genes (*C9orf89*, *CENPA*, *PISD*, and *TRAF2*) were significantly upregulated after exposure to cetuximab compared to parent cells (Figure 5).

### 2.4. Alteration of mRNA Expression of C9orf89, CENPA, PISD, and TRAF2 in OSCC Patients Determined by TCGA-OSCC Analysis

The high expression status of four genes in the TCGA-OSCC cohort using cBioportal is illustrated in Oncoprint. In 321 OSCC clinical samples, we noted high mRNA expression for *C9orf89* in 50%, *CENPA* in 40%, *PISD* in 33%, and *TRAF2* in 22%, and (Z-score ≥ 0). These genes showed significantly higher mRNA expression with increasing DNA copy number (Figure 6A, Appendix A). The expression of all four genes was often increased simultaneously (mRNA Z-score ≥ 0, Figure 6B). These data suggest that the four genes (which do not share the same locus) are regulated by SEs specific to cells that had been exposed to cetuximab for a prolonged period.

Patients with OSCC who had increased expression in at least one of the four genes showed an unfavorable survival outcome and were characterized by aberrant cell cycle gene signatures (Figure 6C,D). On the other hand, the OSCC patients without any increase of the four genes were associated with Focal Adhesion-PI3K-Akt-mTOR-signaling pathway, Ras Signaling, and Chemokine signaling pathway and had a more favorable prognosis (Figure 6D, Appendix A).

This analysis of clinical specimens of TCGA-OSCC suggested that the four candidate genes were co-altered by the formation of a super-enhancer and these changes negatively affected the prognosis.

### 2.5. Immunostaining of C9orf89, CENPA, PISD, and TRAF2 in OSCC Clinical Tissues

Immunohistochemical staining of C9orf89, CENPA, PISD, and TRAF2 was analyzed with the Protein Atlas database (Figure 7, Appendix A). 

C9orf89: Normal tissues displayed occasional nuclear positivity for HPA010921 whereas HPA038297 showed negative staining. In cancer tissues, both antibodies displayed moderate to strong immunoreactivity in the cytoplasm.

CENPA: Weak to strong nuclear positivity was observed in basal cells and parabasal cells in normal tissues. In addition to nuclear positivity, weak cytoplasmic immunoreactivity was observed in malignant cells.

PISD: HPA031090 showed strong cytoplasmic immunoreactivity in both normal and cancer tissues. On the other hand, HPA031091 displayed weak positive staining in normal tissue.

TRAF2: Most normal cells showed negative to weak cytoplasmic positivity with both antibodies, whereas cancer tissues showed strong cytoplasmic positivity.

The moderate to high expression of each gene was confirmed on the cancer tissues, however, normal epithelial tissues were stained in case of a few antibodies.

## 3. Discussion

OSCC is a highly malignant cancer, and the 5-year survival of OSCC has remained below 50% [3,36]. Discovery of drug susceptibility markers and molecules involved in drug resistance is essential for improving the prognosis of patients with OSCC. A vast number of studies have shown that dysregulated epigenetic control of cancer cells is closely involved in malignant transformation, metastasis, and drug resistance [14,15,16].

The super-enhancer concept is critically important in cancer research, and the identification of cancer cell-specific super-enhancers has been vigorously pursued [22,33,34,37]. A recent study using H3K27ac ChiP-seq analysis of HSC4 and BHY cells showed that 41 genes were regulated by super-enhancers [38]. Among these genes, high expression of *AHCY*, *KCMF1*, *MANBAL*, and *TFDP1* predicted poor prognosis of the patients with OSCC [38]. It is evident that genome-wide super-enhancer analyses provide novel information regarding OSCC/HNSCC molecular pathogenesis.

Targeted molecular therapies that specifically block the oncogenic signaling pathways characteristic of cancer cells have improved patient prognosis [39]. Overexpression of EGFR and activation of EGFR-mediated oncogenic pathways are frequently observed in OSCC patients [40,41]. Therefore, anti-EGFR antibodies, such as cetuximab, are used for the treatment of this disease [4,6,42,43]. During treatment with cetuximab, cancer cells acquire genetic alterations such as gain-of-function mutations in *EGFR* and *KRAS*, resulting in treatment resistance [44,45]. In order to explore the molecular mechanism involved in cetuximab resistance, we established cetuximab-resistant OSCC cell lines and performed genome-wide gene expression analysis [35]. In this study, we analyzed super-enhancers involved in cetuximab-resistance.

A total of 64 chromosomal loci were identified as active super-enhancers (SE) by active enhancer marker histone H3 lysine 27 acetylation (H3K27ac) ChIP-Seq. Ultimately, we identified four genes (*C9orf89*, *CENPA*, *PISD*, and *TRAF2*) as super-enhancer-mediated prognostic markers of OSCC patients. In addition, these genes might well be involved in drug resistance in OSCC cells. Clarifying the role of these genes using various drug-resistant cell lines is an important task. Furthermore, it is necessary to confirm whether the expression of these genes fluctuates in clinical specimens before and after drug treatment.

*TRAF2* is a member of the TNF-receptor-associated factor family. It acts as a mediator of TNF-induced signaling [46,47,48]. Overexpression of TRAF2 enhances the malignant phenotype of gastric cancer cells [47]. In nasopharyngeal carcinoma cells, overexpression of *TRAF2* promotes cancer cell proliferation and anchorage-independent growth [48]. Importantly, its overexpression was associated with resistance to irradiation [48].

*CENPA* determines the location of the centromere on chromosomes in mitosis. CENPA protein is a histone H3 variant that replaces one or both of the standard H3 histones in the nucleosome histone complex within the centromere [49]. Expression levels of *CENPA* are associated with patients’ responses to chemo- and radiotherapy and they predict poorer survival rates of cancer patients [50,51].

Aberrant expression of *NRMT* (N-terminal regulator of chromatin condensation 1 methyltransferase) has been reported in various cancers [52,53]. *NRMT* controls the expression of *CENPA* through a promoter region of *CENPA* [54]. Furthermore, CENPA induces the transcription of Myc and elevates the expression of Bcl2 in retinoblastoma cells [53]. Importantly, aberrant expression of the NRTM/CENPA/Bcl2 axis developed in cisplatin (CDDP) resistance of retinoblastoma cells [53].

Interestingly, the group characterized by high expression of any of the four genes (the high expression group) was enriched for pathway terms associated with cell cycle upregulation in GSEA analysis compared to the group without such expression (low expression group). In this “high expression” patient population, combination therapies targeting the cell cycle, such as CDK4/6 inhibitors, are expected to provide responses not achieved with cetuximab alone. In fact, combination therapy with the CDK4/6 inhibitor palbociclib and cetuximab was used in a phase II trial, and the response rate in the cetuximab-resistant group was 19% (5 of 32 patients had PR) [55].

Pathways that were enriched in the “low-expression” group included the Focal Adhesion-PI3K-Akt-mTOR-signaling pathway, Ras Signaling and the chemokine signaling pathway. These results suggest approaches that co-inhibit RAS-MAPK-ERK signaling and PI3K-AKT-mTOR signaling, or that combine cetuximab with immune checkpoint inhibition. Such approaches have begun to be investigated in preclinical and clinical trials, and co-inhibition of RAS-MAPK-ERK and PI3K-AKT-mTOR signaling has demonstrated antitumor effects in the HNSCC PDX model [56] and the PIK3CA+ OSCC preclinical model [57]. Clinical efficacy will be confirmed in the KURRENT study (NCT04997902).

A combination of immune checkpoint inhibition with cetuximab has shown results suggesting efficacy as measured by CXCL10 expression at the ex vivo assay level [58]. A combination of pembrolizumab and cetuximab in a recent phase II trial demonstrated promising responses and a manageable safety profile [59].

We believe that the four genes we identified in this analysis will help elucidate the mechanisms underlying drug-resistance, including cetuximab-resistance in OSCC cells. In addition, it may be possible to construct a drug susceptibility/diagnostic system for OSCC patients to enable stratification into appropriate treatment regimens based on the expression of those genes as an index. Finally, we acknowledge a limitation of the current approach. That is, the TCGA-OSCC cohort we analyzed in this study does not provide direct evidence based on gene expression profiling of cetuximab-resistant patients and is therefore speculative. However, we anticipate validating these results in ongoing clinical trials.

## 4. Materials and Methods

### 4.1. Parental Cell Lines and Cetuximab Long-Term Exposure Cell Lines

OSCC-derived cell lines (HSC-3 and SAS) were purchased from and authenticated by the Human Science Research Resources Bank (Osaka, Japan) or the RIKEN Bio Resource Center (Ibaraki, Japan) and cultured as previously described [60]. Parental cell lines were subjected to prolonged exposure to cetuximab as described previously [35] and used as cetuximab long-term exposure cell lines (Cmab-LTE HSC-3 and Cmab-LTE SAS). Genomic alterations (EGFR mutation status (exon 18 [G719X], exon 19 [E746_A750 deletion], exon 20 [V769_V774 insertions], exon 20 [T790M], and exon 21 [L858R]) and KRAS (codon 12/13) genes) in these cell lines were previously assessed and were not detected in all cell lines [35].

### 4.2. H3K27ac CHIP Sequencing

For H3K27ac ChIP-seq and super-enhancers analysis, cells were fixed with 1% formaldehyde for 15 min and quenched with 0.125 M glycine, and the frozen cell pellet containing 1 × 10^7^ cells was sent to Active Motif Inc. (Carlsbad, CA, USA) according to the manufacturers’ instructions. ChIP-seq analysis was performed by Active Motif Inc. as reported [61,62,63].

Briefly, chromatin was isolated after treatment with Chromatin Prep Lysis Buffer (Active Motif) containing non-ionic detergent and protease inhibitors, followed by disruption with a Dounce homogenizer. Genomic DNA was sheared to an average length of 300–500 bp using an EpiShear probe sonicator (Active Motif, cat# 53051) and a cooled sonication platform (Active Motif, cat# 53080). The segments of interest were immunoprecipitated by 4 μL of specific antibody against H3K27Ac (Active Motif, cat# 39133, Lot 16119013). The protein and DNA complexes were washed, eluted from the Agarose beads and were treated with SDS buffer, RNase, and proteinase K. Crosslinks were reversed and ChIP DNA was purified by phenolchloroform extraction and ethanol precipitation.

Sequencing libraries were prepared and sequenced on Illumina’s NextSeq 500 (75 nt reads, single-end). The reads were aligned to the human genome (hg38) by BWA (default settings) and non-duplicated mapped reads (mapping quality > 25) were used for further analysis. Alignments were extended in silico at their 3′-ends to a length of 200 bp and assigned to 32-nt bins along the genome. The histograms were stored in bigWig files and peak locations were determined using the MACS algorithm (v2.1.0) with a cutoff *p*-value = 1 × 10^−7^. ENCODE blacklist, known as false ChIP-Seq peaks, were removed. Signal maps and peak locations were used as input data to an Active Motifs proprietary analysis program.

Our data resulting from CHIP Sequencing analysis were deposited in the GEO data-base (accession number: GSE205455).

### 4.3. Super Enhancer Analysis

Super Enhancer regions were determined using BED Tools software [64] and standard UNIX commands. The first step is identical to ROSE and stitches together MACS2 peak regions that are less than 12.5 kb apart (stitching parameter = 12.5 kb). Next, the number of tags (aligned reads in the normalized, BAM-derived BED files) in each of the stitched regions was determined, and the tag numbers were then used to rank the regions. The top 5% were designated Super Enhancers and those were annotated with genes and promoters.

### 4.4. RNA Extraction and Quantitative Reverse-Transcription PCR (qRT-PCR)

Total RNA was isolated using TRIzol reagent and the PureLink™ RNA Mini Kit (Invitrogen/Thermo Fisher Scientific (Waltham, MA, USA)). Reverse transcription was achieved with the High Capacity cDNA Reverse Transcription Kit (Applied Biosystems, Waltham, MA, USA). qRT-PCR was performed with SYBR Green using the StepOnePlus™ Real-Time PCR System (Applied Biosystems). *GAPDH* was used as the normalized control. Primer sequences used in this report are shown in Appendix A.

### 4.5. Analysis of Expression and Clinical Significance of Candidate Gene Expression in OSCC by TCGA Database Analysis

Analysis of expression and clinical significance of the candidate genes was performed by using data from cBioPortal (http://cbioportal.org), accessed on 10 April 2020 [65]. TCGA-OSCC data were defined as TCGA-HNSC data (Firehose Legacy) in which the primary site was the tongue, oral cavity, the floor of the mouth, buccal mucosa, alveolar ridge, hard palate, or lip. We ran queries on four genes (*C9orf89*, *CENPA*, *PISD*, and *TRAF2*) to specify changes in mRNA expression (Z score ≥ 0) and analyzed mutual exclusivity and overall survival. The ranked gene list for gene set enrichment analysis (GSEA) was obtained from the comparison of mRNA expression levels between altered and non-altered groups and uploaded into WEB-based GEne SeT AnaLysis Toolkit, “WebGestalt” (http://www.webgestalt.org), accessed on 10 April 2022 [66]. We applied “Wikipathway cancer” dataset for GSEA (https://www.wikipathways.org/index.php/WikiPathways), accessed on 10 April 2022 [67].

### 4.6. Immunostaining Analysis by Protein Atlas Database

To confirm the protein expression levels of target genes, images of immunohistochemical staining were downloaded from The Human Protein Atlas database (https://www.proteinatlas.org), accessed on 24 March 2022 [68,69]. The Human Protein Atlas is a Swedish-based program started in 2003 with the purpose of mapping all human proteins. All the data exhibited in this program is open access for exploration of the human proteome.

The links to the information of each gene, clinical features of the HNSCC patients and the antibody information is summarized in Appendix A. 

### 4.7. Statistical Analysis

JMP Pro 15 (SAS Institute Inc., Cary, NC, USA) was used for statistical analyses. Comparisons between the two groups were assessed by Welch’s *t*-test. One-way analysis of variance (ANOVA) was used for comparisons between multiple groups. Overall survival analysis were analyzed by log-rank test. A *p*-value < 0.05 was considered statistically significant. Quantitative data are presented as the means and standard errors.

## 5. Conclusions

Elucidation of the mechanism underlying cells’ resistance to anti-cancer drugs is critical for improving the prognosis of OSCC patients. In this study, H3K27ac ChIP-Seq was applied to investigate cetuximab treatment-induced genomic changes. A total of 64 SE peaks were detected in OSCC cells following long-term exposure to cetuximab. A total of 131 genes were involved in the SE region, of which four genes (*C9orf89*, *CENPA*, *PISD*, and *TRAF2*) affected the prognosis of patients with OSCC. Analysis of these genes will contribute to improved understanding of drug resistance in OSCC patients.

## Figures and Tables

**Figure 1 ijms-23-09154-f001:**
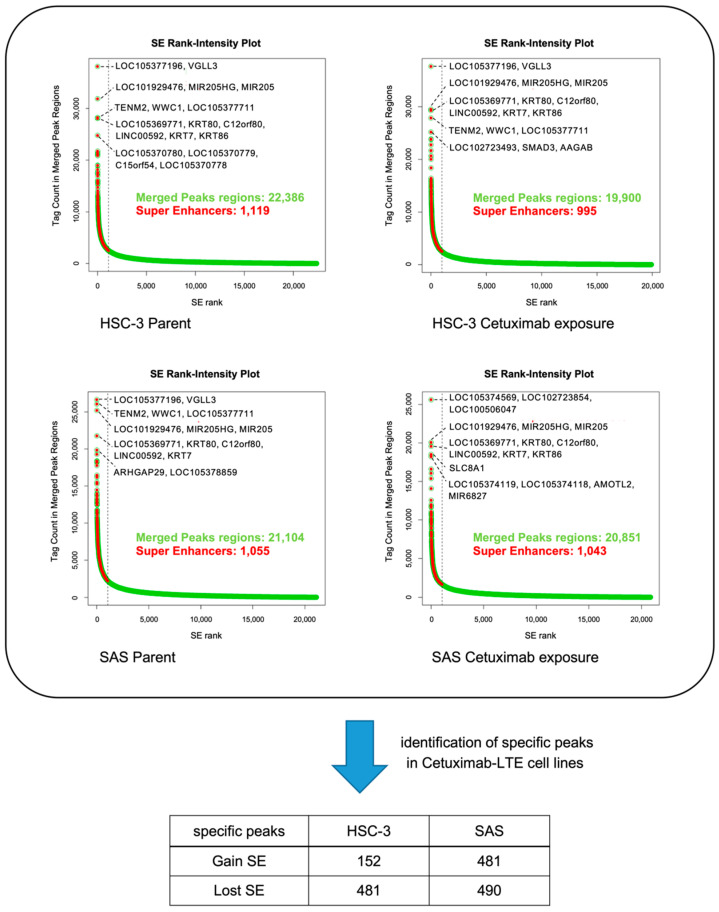
Detection of SE peaks in OSCC cell lines after long-term exposure to cetuximab (Cmab-LTE). Identification of SEs in parental cell lines (HSC-3 and SAS) and Cmab-LTE cell lines (HSC-3 Cmab-LTE and SAS Cmab-LTE). The proximal genes of the top 5 super-enhancers in parental and Cmab-LTE cell lines are marked. Identification of specific gained and lost SE peaks in Cmab-LTE cell lines.

**Figure 2 ijms-23-09154-f002:**
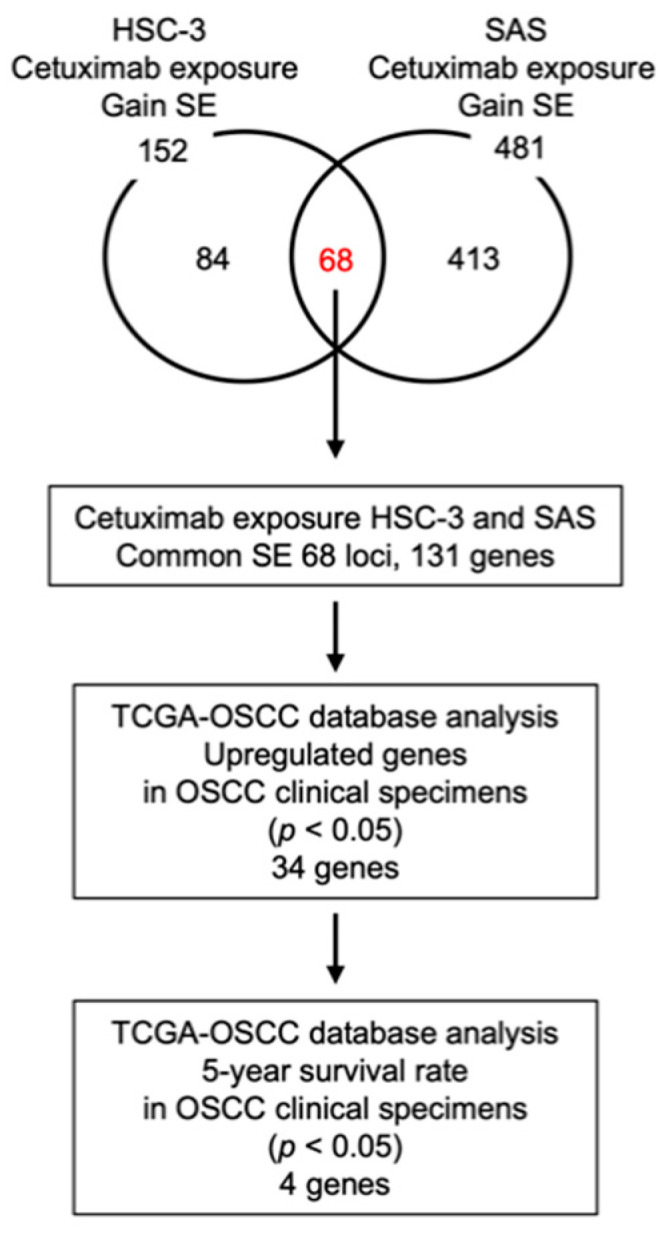
Flowchart of methods used for the identification of prognostic genes in OSCC patients. Venn diagram shows the overlapped gain in SE peak numbers between HSC-3 Cmab-LTE and SAS Cmab-LTE cell lines. A total of 68 SE loci and 131 corresponding genes in SE regions are identified. TCGA-OSCC database analysis shows that (*C9orf89*, *CENPA*, *PISD*, and *TRAF2*) are closely involved in OSCC molecular pathogenesis.

**Figure 3 ijms-23-09154-f003:**
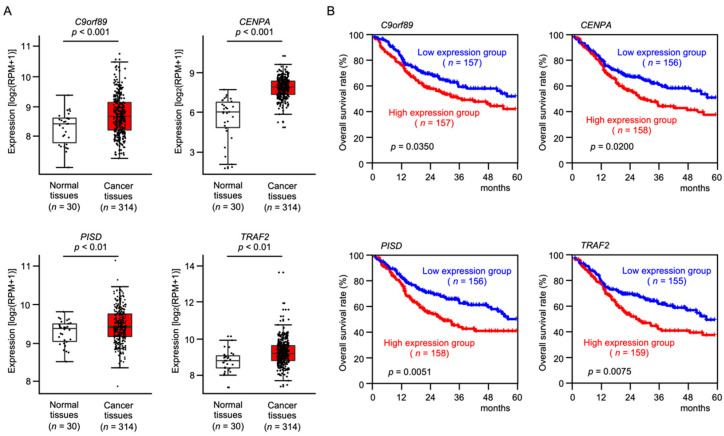
Clinical significance of 4 genes (*C9orf89*, *CENPA*, *PISD*, and *TRAF2*) in OSCC clinical specimens determined by TCGA-OSCC analysis. (**A**) Expression levels of 4 target genes (*C9orf89*, *CENPA*, *PISD*, and *TRAF2*) in OSCC clinical specimens from TCGA-OSCC. All genes were found to be upregulated in OSCC tissues (*n* = 314) compared with normal tissues (*n* = 30). (**B**) Clinical significance of four target genes (*C9orf89*, *CENPA*, *PISD* and *TRAF2*) according to TCGA-OSCC data analysis. Kaplan–Meier curves of the 5-year overall survival rates according to the expression of each gene. Patients were divided into 2 groups according to the median gene expression level: high and low expression groups. The red and blue lines represent the high and low expression groups, respectively. High expression levels of all 4 genes significantly predicted a poorer prognosis in patients with OSCC. Nominal *p*-value was calculated by log-rank test.

**Figure 4 ijms-23-09154-f004:**
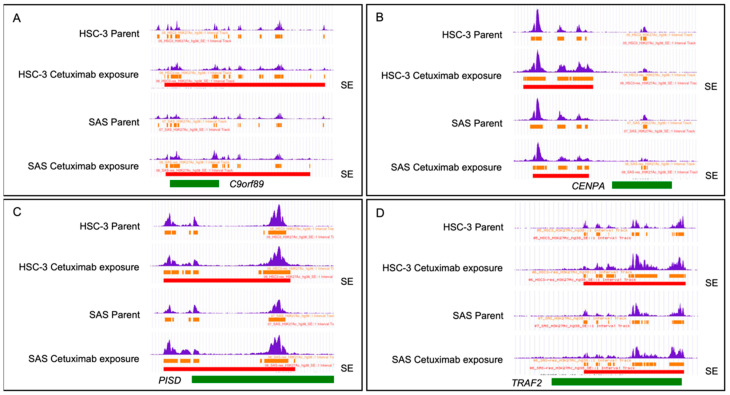
The genome browser view show the H3K27ac signals in OSCC cell lines. (**A**–**D**) Specific H3K27ac signals in Cmab-LTE cell lines are shown in 4 chromosomal regions located around the *C9orf89*, *CENPA*, *PISD*, and *TRAF2* genes. SEs characterized by H3K27ac following cetuximab exposure in HSC-3 and SAS lines based on the UCSC genome browser. MACS-peak regions/intervals are represented in orange bars and are stitched together to generate SEs if their distance is <12.5 kb. The top 5% stitched regions are designated SEs and shown by red bars. (**A**) *C9orf89* (chr9:93,096,217–93,113,283), (**B**) *CENPA* (chr2:26,786,014–26,794,589), (**C**) *PISD* (chr22:31,618,491–31,662,564), and (**D**) *TRAF2* (chr9:136,881,933–136,926,621) loci are indicated in green bars. The blue peaks in the top track are the bigWIG data.

**Figure 5 ijms-23-09154-f005:**
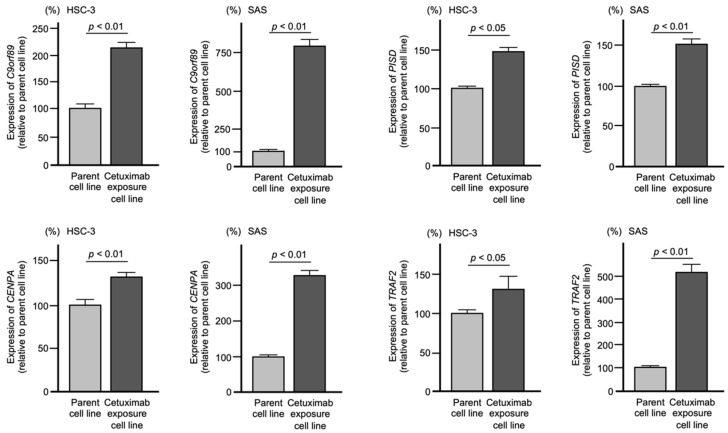
Expression levels of 4 genes (*C9orf89*, *CENPA*, *PISD*, and *TRAF2*) in OSCC cell lines after prolonged cetuximab exposure. The expression levels of 4 genes (*C9orf89*, *CENPA*, *PISD*, and *TRAF2*) were increased by the cetuximab treatment compared with the parental cells. Gene expression was measured by SYBR Green Real-time PCR methods. *GAPDH* was used as an internal control.

**Figure 6 ijms-23-09154-f006:**
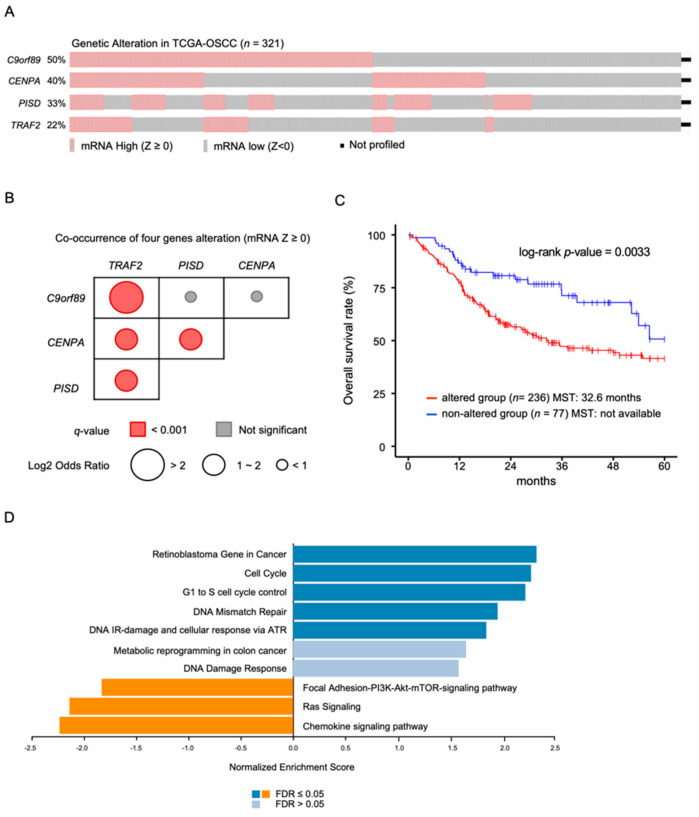
Alterations of mRNA expression levels of 4 genes (*C9orf89*, *CENPA*, *PISD* and *TRAF2*) in OSCC clinical specimens based on TCGA-OSCC analyses. (**A**) Oncoprint of TCGA-OSCC on cBioPortal filtered by the mRNA expression (Z score ≥ 0) from the query for 4 genes. (**B**) Mutual exclusivity of 4 genes. *q*-value was derived from the Benjamini–Hochberg FDR correction procedure. Odds ratio shows how strongly the presence or absence of alterations in one are associated with the presence or absence of alterations in another in the selected samples. (**C**) Kaplan–Meier curves of overall survivals between altered and non-altered group (MST: median survival time). (**D**) Bar chart and enrichment plots of GSEA of alteration of 4 genes (FDR: false discovery rate).

**Figure 7 ijms-23-09154-f007:**
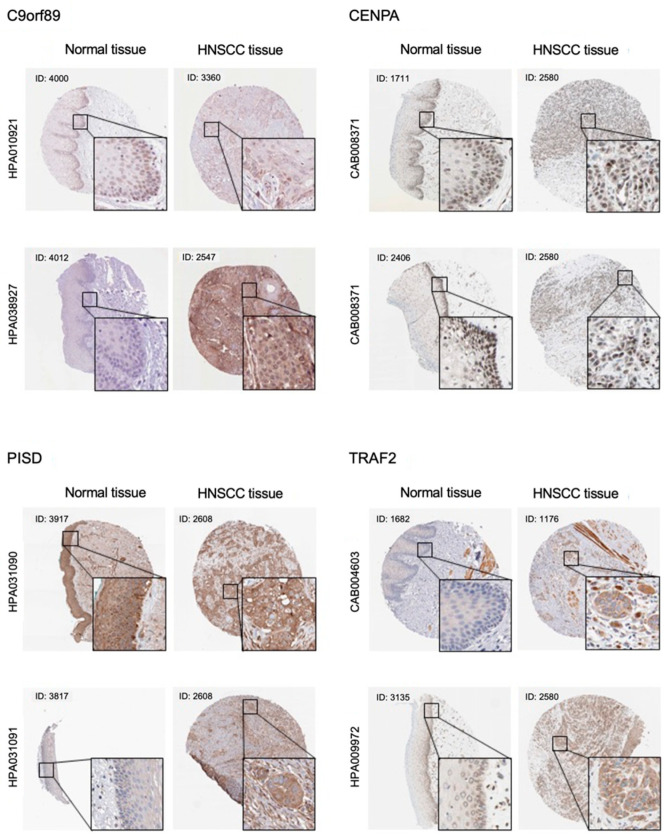
Protein expression of C9orf89, CENPA, PISD, and TRAF2 in OSCC clinical specimens according to the Protein Atlas database. Protein expression of C9orf89, CENPA, PISD and TRAF2 in OSCC clinical specimens is based on the Protein Atlas database. C9orf89: Both antibodies showed negative staining in normal tissues. In cancer tissues, HPA010921 showed weak immunoreactivity in the cytoplasm whereas HPA038297 showed strong immunoreactivity. CENPA: CAB008371 displayed strong nucleic positivity in both normal and cancer tissues, whereas weak cytoplasmic positivity was shown in both cancer tissues. PISD: Both antibodies strongly stained cancer cytoplasm whereas HPA031090 stained normal epithelium as well. TRAF2: Both antibodies stained cancer cytoplasm strongly whereas HPA0099972 weakly stained basal cells in normal epithelium.

**Table 1 ijms-23-09154-t001:** The detailed information of the genes in the super-enhancer regions.

No.	Merged Region	Chromosome	Start	End	Length	Gene List	Position
1	3	1	3,453,352	3,483,697	30,345	*PRDM16*, *ARHGEF16*	downstream, in gene
2	35	1	31,689,438	31,715,640	26,202	***COL16A1***, *LOC101929444*, ***BAI(ADGRB2)***	in gene, downstream, downstream
3	52	1	46,170,382	46,203,757	33,375	*PIK3R3*, *LOC105378695*, *TSPAN1*, ***POMGNT1***, *LURAP1*	upstream, upstream, in gene, downstream, upstream
4	89	1	151,536,650	151,553,908	17,258	*CGN*, *TUFT1*, *MIR554*	downstream, in gene, upstream
5	121	1	180,501,359	180,533,622	32,263	** *ACBD6* **	upstream
6	131	1	200,885,778	200,903,119	17,341	*GPR25*, ***C1orf106***, *MROH3P*	downstream, in gene, upstream
7	135	1	202,566,931	202,600,283	33,352	*PPP1R12B*, *SYT2*, *LOC105371686*, *LOC105371685*	in gene, downstream, upstream, upstream
8	166	1	240,758,392	240,811,761	53,369	*LOC100506929*, *RGS7*, *LOC105373229*	upstream, in gene, upstream
9	208	10	72,243,611	72,278,769	35,158	*ANAPC16*	downstream
10	212	10	75,207,300	75,264,014	56,714	*VDAC2*, *COMTD1*	downstream, in gene
11	248	10	132,395,732	132,422,986	27,254	*LRRC27*, *PWWP2B*, *LOC105378568*, ***C10orf91***	downstream, in gene, downstream, upstream
12	254	11	8,806,999	8,841,147	34,148	*ST5*, *LOC102724784*, *RNA5SP330*	in gene, downstream, upstream
13	259	11	12,788,054	12,842,696	54,642	*TEAD1*	in gene
14	284	11	63,559,618	63,584,348	24,730	*RARRES3*, ***HRASLS2***, *PLA2G16*, *LOC105369335*	downstream, upstream, downstream, upstream
15	289	11	65,371,027	65,392,675	21,648	***TIGD3***, *SLC25A45*	downstream, in gene
16	315	11	114,280,212	114,309,047	28,835	*NNMT*	upstream
17	390	12	47,811,914	47,836,614	24,700	*LOC105369749*	upstream
18	429	12	79,545,485	79,567,438	21,953	*PAWR*	downstream
19	460	12	122,696,516	122,727,563	31,047	*HCAR2*, *HCAR3*, *HCAR1*	upstream, downstream, downstream
20	471	13	33,118,131	33,129,383	11,252	** *STARD13* **	in gene
21	482	13	79,480,690	79,494,906	14,216	*NDFIP2-AS1*, *NDFIP2*	upstream, in gene
22	502	14	22,588,162	22,622,517	34,355	*DAD1*, *ABHD4*	upstream, in gene
23	585	15	73,973,341	73,997,593	24,252	*STOML1*, ***PML***	in gene, upstream
24	633	16	68,731,540	68,802,326	70,786	*CDH1*	downstream
25	641	16	81,559,343	81,602,113	42,770	*MIR6504*	in gene, upstream
26	672	17	17,900,207	17,972,359	72,152	*TOM1L2*, *LRRC48*, *ATPAF2*	in gene, upstream, downstream
27	674	17	19,706,613	19,729,962	23,349	***SLC47A2***, *ALDH3A1*	in gene, downstream
28	699	17	42,658,915	42,683,176	24,261	*HMGB3P27*, ***TUBG2***, *PLEKHH3*, *CCR10*, ***CNTNAP1***, *EZH1*, *MIR6780A*	downstream, downstream, in gene, downstream, upstream, downstream, downstream
29	746	17	82,096,030	82,108,208	12,178	*FASN*	upstream
30	764	18	57,770,620	57,846,982	76,362	*ATP8B1*, *LOC1 + G3305376870*, *RSL24D1P11*	upstream, downstream, upstream
31	775	19	2,523,762	2,555,593	31,831	*LOC101929097*, *GNG7*	upstream, in gene
32	776	19	4,367,944	4,403,867	35,923	*MPND*, *SH3GL1*, ***CHAF1A***	downstream, in gene, upstream
33	778	19	6,719,422	6,747,512	28,090	*C3*, *GPR108*, *MIR6791*, *TRIP10*, ***SH2D3A***	upstream, in gene, downstream, upstream, downstream
34	790	19	18,361,769	18,388,017	26,248	*PGPEP1*, *GDF15*, *MIR3189*, *LRRC25*	downstream, upstream, upstream, downstream
35	797	19	38,251,595	38,320,090	68,495	*PPP1R14A*, *SPINT2*, ***YIF1B***, *C19orf33*, *KCNK6*	upstream, in gene, downstream, upstream, upstream
36	811	19	43,104,469	43,131,735	27,266	***PSG5***, ***PSG2***	in gene, upstream
37	817	19	46,191,369	46,232,922	41,553	***IGFL2***, *LOC105372424*, *LOC645553*, *LOC105372423*, *LOC105372422*, *IGFL1*	downstream, upstream, in gene, downstream, downstream, upstream
38	840	2	26,755,019	26,773,104	18,085	***C2orf18(SLC35F6)***, ***CENPA***	upstream, upstream
39	849	2	36,476,174	36,505,315	29,141	*CRIM*	in gene
40	905	2	85,237,328	85,298,810	61,482	*TCF7L1*, *LOC102724579*, *LOC105374839*	in gene, downstream, downstream
41	995	20	10,653,952	10,675,733	21,781	***JAG1***, *MIR6870*, *LOC105372526*	in gene, upstream, upstream
42	998	20	19,903,485	19,958,418	54,933	*RIN2*	in gene
43	1081	21	38,898,236	38,926,384	28,148	*LOC400867*	in gene
44	1084	21	41,751,747	41,787,733	35,986	*RIPK4*, *MIR6814*, *LOC102724800*, *PRDM15*	upstream, upstream, in gene, downstream
45	1101	22	24,950,104	24,997,098	46,994	*TMEM211*, *KIAA1671*	upstream, in gene
46	1111	22	31,629,899	31,663,559	33,660	*SFI1*, ***PISD***, *MIR7109*, *PRR14L*	downstream, in gene, upstream, downstream
47	1121	22	37,887,250	37,908,095	20,845	*EIF3L*, ***MICALL1***	downstream, upstream
48	1127	22	40,482,368	40,542,411	60,043	***MKL1***, *LOC101927257*, *LOC105373037*	in gene, upstream, upstream
49	1135	22	46,731,873	46,775,788	43,915	*CERK*, *LOC105373077*, *TBC1D22A*	upstream, upstream, upstream
50	1155	3	37,934,276	37,947,923	13,647	*CTDSPL*, *MIR26A1*	in gene, upstream
51	1189	3	123,583,773	123,653,831	70,058	*HACD2*, *MYLK-AS1*	upstream, in gene
52	1207	3	153,130,215	153,165,195	34,980	** *RAP2B* **	upstream
53	1223	3	183,253,290	183,297,852	44,562	*MCF2L2*, ***B3GNT5***, *RNA5SP151*	in gene, downstream, upstream
54	1237	3	197,482,394	197,521,067	38,673	*LOC105374308*, *LOC105374309*, *BDH1*	upstream, downstream, downstream
55	1299	5	57,681,786	57,700,891	19,105	*LOC101928505*	downstream
56	1392	6	33,731,250	33,789,203	57,953	***C6orf125(UQCC2)***, *IP6K3*, ***LEMD2***, *LOC105375024*, *MLN*	upstream, upstream, downstream, upstream, downstream
57	1473	7	27,080,276	27,115,997	35,721	***HOXA1***, *HOTAIRM1*, *HOXA2*, *LOC105375205*	upstream, in gene, downstream, upstream
58	1475	7	28,034,605	28,067,086	32,481	*JAZF1*, *LOC105375208*	in gene, in gene
59	1496	7	47,633,320	47,694,065	60,745	*LINC01447*, ***C7orf65***	downstream, downstream
60	1555	8	22,561,116	22,605,497	44,381	*PPP3CC*, *SORBS3*, *LOC105379320*, *PDLIM2*, ***C8orf58***, *CCAR2*, *BIN3*	downstream, downstream, upstream, in gene, upstream, upstream, downstream
61	1634	8	140,722,495	140,734,727	12,232	*MIR151A*	downstream
62	1640	8	142,777,613	142,796,025	18,412	*LYNX1*, *LY6D*	upstream, upstream
63	1655	9	22,079,706	22,119,693	39,987	*CDKN2B-AS1*	in gene
64	1672	9	93,093,990	93,149,539	55,549	*SUSD3*, *LOC101927993*, ***C9orf89***, *NINJ1*, *LOC105376150*	downstream, upstream, downstream, in gene, upstream
65	1677	9	106,860,435	106,921,639	61,204	*LOC105376204*, *ZNF462*	upstream, in gene
66	1703	9	129,314,856	129,336,831	21,975	*C9orf106*, *LINC01503*	downstream, upstream
67	1708	9	136,533,550	136,579,457	45,907	*NOTCH1*, *MIR4673*, *LOC1053763204*, *MIR4674*, *LINC01573*	upstream, upstream, downstream, upstream, downstream
68	1709	9	136,881,365	136,905,108	23,743	*MAMDC4*, *EDF1*, *LOC105376326*, ***TRAF2***, *MIR4479*	downstream, upstream, upstream, in gene, downstream

## Data Availability

Our data resulting from CHIP Sequencing analysis were deposited in the GEO database (accession number: GSE205455).

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
