# Peer review of "Genome-Wide Super-Enhancer-Based Analysis: Identification of Prognostic Genes in Oral Squamous Cell Carcinoma"

_ijms, 2022, doi:10.3390/ijms23169154_

Round 1

Reviewer 1 Report

Reviewer's report

Title: Genome-Wide Super-Enhancer-Based Analysis: Identification of Prognostic Genes in Oral Squamous Cell Carcinoma

Version: 1 Date:  2022 July 4th

Reviewer's report:

In this manuscript, the Authors investigated by ChIP-SEQ for H3K27 active enhancers in OSCC cell lines before and after treatment with cetuximab for prolonged period. They found 64 active chromosomal loci associated with 131 genes, of which 34 were found to be up-regulated in OSCC tissues analyzing TCGA database. Among these ones, 4 (C9orf89, CENPA, TRAF2, PISD) displayed good correlation between their high gene expression and poor prognosis.

Overall, this work is well written and the findings outlined in this manuscript are very interesting.

Major Revisions

It would have been useful to check a series of clinical cases (at least 50-100 cases) by RTqPCR to verify the prognostic potential of these 4 biomarkers linked to cetuximab treatment in a real field. At present, this paper lacks the final part related to the clinical validation and its utility.

Author Response

Revise letter (Manuscript ID: ijms-1802581)

Special Issue "Advances in Molecular Mechanism of Head and Neck Cancer"

July 20, 2022

Special Issue Editor

Dr. Nijiro Nohata

Dear Dr. Nohata

We would like to express our gratitude for your consideration of our above-mentioned manuscript for publication in IJMS. Enclosed, please find the revised manuscript (Manuscript ID: ijms-1802581) along with a detailed explanation of the revisions, which were made based on the reviewers’ comments. All changes are highlighted in the revised manuscript.

Reviewer #1

Reviewer's report:

In this manuscript, the Authors investigated by ChIP-SEQ for H3K27 active enhancers in OSCC cell lines before and after treatment with cetuximab for prolonged period. They found 64 active chromosomal loci associated with 131 genes, of which 34 were found to be up-regulated in OSCC tissues analyzing TCGA database. Among these ones, 4 (C9orf89, CENPA, TRAF2, PISD) displayed good correlation between their high gene expression and poor prognosis.

Overall, this work is well written and the findings outlined in this manuscript are very interesting.

Major Revisions:

It would have been useful to check a series of clinical cases (at least 50-100 cases) by RTqPCR to verify the prognostic potential of these 4 biomarkers linked to cetuximab treatment in a real field. At present, this paper lacks the final part related to the clinical validation and its utility.

Response:

We are fully aware of the comments you have pointed out.

It is indispensable for the development of this study to verify with clinical specimens whether the genes searched in this study are involved in drug resistance. However, obtaining clinical samples after cetuximab or anticancer drug treatment has clinical and ethical problems and is difficult to analyze.

The following text was added (Discussion 5th chapter) to the reviewer's suggestions.

Clarifying the role of these genes using various drug-resistant cell lines is an important task. Furthermore, it is necessary to confirm whether the expression of these genes fluctuates in clinical specimens before and after drug treatment.

Thank you for your constructive comments and suggestions. We believe that our manuscript has been greatly improved and is now suitable for publication in IJMS. Again, thank you for your consideration of our manuscript for publication in your journal.

Sincerely yours,

Naohiko Seki, Ph.D.

Department of Functional Genomics

Chiba University Graduate School of Medicine

1-8-1 Inohana, Chuo-ku,

Chiba 260-8670, Japan

Phone: +81-43-226-2971

Fax: +81-43-227-3442

e-mail: naoseki@faculty.chiba-u.jp

Reviewer 2 Report

Dear Authors,

This is an interesting in vitro study investigating the role of super-enhancer regions of the mammalian genome in the response to treatment with cetuximab.

In summary, the introduction is clear and well written. However, the Material and Methods section lacks important information to reproduce the study. Concerning section 2.3 and 2.5 of the results, I have a feeling that this is not very robust data, as the analysis itself is based on results obtained from only two cell lines and the analyzed online-datasets are inconsistent and difficult to reconstruct. The discussion somewhat fails to consider its own findings more in relation to the literature

To improve the manuscript, the following points should also be considered.

Does the section organization match the style of the journal (?): 3. Discussion > 4. Materials and Methods > 5. Conclusions

Figure 3: please provide survival rate in %. (I would prefer to specify the time in years or in intervals of 12 months, but this is only a small thing). (please also check Figure 5 for consistent presentation)

Figure 4: requires sub-labeling of individual sub-figures.

MM:

The terms “General” and “specific cell lines” are misleading (e.g. parental and cetuximab-treated cell lines would be more specific)

Lines 313-320 requires revision for clarity: “and cultured” => under which conditions(?); “(Cmab LTE SAS, and Cmab-LTE HSC-3)” => are these derived from Sa3, and HSC-3? => this is not recognizable due to the current nomenclature; Cmab-LTE/ Cmab LTE are these abbreviations (please write out abbreviations at first time using in the entire manuscript – e.g. what us BWA?)? “Genomic alterations…” is this used for authentication (?) => move next to the authentication issue.

Line 324: Mentioning companies usually requires more information than just the name => please check the journal's guidelines in this regard. In addition, the procedure from cultivation to shipment requires further information or reference to a citable protocol to reproduce the study. EpiShear probe sonicator (cat# 53051): which company? Please indicate which experiments and/or data analyses of the study were performed by a/which commercial institution(s).

Please indicate at which steps biological and/or technical replicates were performed?

Lines 328-332: requires more detailed information about procedures (e.g. volumes, type of beads..) and buffer compositions (SDS-buffer (?)) or respective references.

Lines 372-373: how can protein expression levels of target genes be confirmed by downloading of images?

Results

Lines 101-104: redundant with next sentence.

Lines 116-117: “Cmab-LTE cell-specific” term seems to me overinterpreted => better change to “…gain peaks were identified in Cmab-LTE HSC-3 and Cmab-LTE SAS compared with the respective parental cell line.”

In my opinion, Table 1 and 2 contain too much detailed information for the main text of the manuscript. On the other hand, both contain some interesting data. I suggest merging Table 2 with Table 1 if possible, since there are some redundancies. E.g., by adding each row from Table 2 (in bold) below the corresponding rows in Table 1. Alternatively, the highly regulated genes from Table 2 could be highlighted in Table 1 (and Table 2 moved to the supplements).

Line 145: move sentence two sentences down

Section 2.4: I propose to substitute this section for section 2.3, as it represents the experimental validation of the upregulation in cell lines shown before.

Section 2.3 and 2.5: I have a feeling that this is not very robust data, as the analysis itself is based on results obtained from only two cell lines. Regarding the selection of samples/patients, the analyzed datasets are inconsistent, difficult to reconstruct, and in particular the cancer patients included in the Cancer Genome Atlas have been selected based on "convenience of collection". How many pictures were available and evaluated for this work? How were pictures selected for presentation?

Section 2.5: “C9orf89: Normal tissues displayed occasional nuclear positivity for HPA010921 whereas HPA038297 showed negative staining.” and “PISD: HPA031090 showed strong cytoplasmic immunoreactivity in both normal and cancer tissues. On the other hand, HPA031091 displayed weak positive staining in normal tissue.” => what is the result (of this study) here, except that two antibodies show different staining? How have antibodies selected for the investigation in this study?

Line 172-174, Figure 5C: “patients with increased expression of the 4 genes” => these are roughly n=15-20 according to “mRNA High” for all 4 genes in Figure 5A but n=236 in Figure 5C. It is unclear how this group of patients is characterized.

Figure 6: Western blot and/or immunostaining of cell line samples would clearly substantiate the mRNA expression data.

Discussion:

Lines 225-231 may fit better / or be redundant with introduction – please check.

Lines 236-249: How does this relate to the results of this work?

Line 291-293: “These results suggest approaches that co-inhibit RAS-MAPK-ERK signaling and PI3K-AKT-mTOR signaling, or that combine cetuximab with immune checkpoint inhibition.” I wonder if the authors have thought about the possibility of testing this in some way in their cell line model.

Author Response

Revise letter (Manuscript ID: ijms-1802581)

Special Issue "Advances in Molecular Mechanism of Head and Neck Cancer"

July 20, 2022

Special Issue Editor

Dr. Nijiro Nohata

Dear Dr. Nohata

We would like to express our gratitude for your consideration of our above-mentioned manuscript for publication in IJMS. Enclosed, please find the revised manuscript (Manuscript ID: ijms-1802581) along with a detailed explanation of the revisions, which were made based on the reviewers’ comments. All changes are highlighted in the revised manuscript.

Reviewer #2

This is an interesting in vitro study investigating the role of super-enhancer regions of the mammalian genome in the response to treatment with cetuximab.

In summary, the introduction is clear and well written. However, the Material and Methods section lacks important information to reproduce the study. Concerning section 2.3 and 2.5 of the results, I have a feeling that this is not very robust data, as the analysis itself is based on results obtained from only two cell lines and the analyzed online-datasets are inconsistent and difficult to reconstruct. The discussion somewhat fails to consider its own findings more in relation to the literature

To improve the manuscript, the following points should also be considered.

Comment:

Does the section organization match the style of the journal (?): 3. Discussion > 4. Materials and Methods > 5. Conclusions

Figure 3: please provide survival rate in %. (I would prefer to specify the time in years or in intervals of 12 months, but this is only a small thing). (please also check Figure 5 for consistent presentation)

Response:

I have confirmed the article style of your journal "IJMS".

The figures have been revised according to the reviewer's comment.

Comment:

Figure 4: requires sub-labeling of individual sub-figures.

Response:

The figure has been revised according to the reviewer's comment.

MM:

Comment:

The terms “General” and “specific cell lines” are misleading (e.g. parental and cetuximab-treated cell lines would be more specific)

Response:

As suggested by the reviewer’s comment, I modified it as follows.

4.1. Parental cell lines and cetuximab long-term exposure cell lines

Comment:

Lines 313-320 requires revision for clarity: “and cultured” => under which conditions(?); “(Cmab LTE SAS, and Cmab-LTE HSC-3)” => are these derived from Sa3, and HSC-3? => this is not recognizable due to the current nomenclature; Cmab-LTE/ Cmab LTE are these abbreviations (please write out abbreviations at first time using in the entire manuscript – e.g. what us BWA?)?

Response:

I thank the reviewer for your suggestions. As suggested by the reviewer’s comment, I modified it as follows.

OSCC-derived cell lines (HSC-3 and SAS) were purchased from and authenticated by the Human Science Research Resources Bank (Osaka, Japan) or the RIKEN Bio Resource Center (Ibaraki, Japan) and cultured as previously described [60].  Parental cell lines were subjected to prolonged exposure to cetuximab as described previously [35] and used as cetuximab long-term exposure cell lines (Cmab-LTE HSC-3 and Cmab-LTE SAS).

Comment:

“Genomic alterations…” is this used for authentication (?) => move next to the authentication issue.

Response:

Genomic alterations were previously investigated to confirm that EGFR and KRAS mutations have not been observed in cetuximab-resistant OSCC cells (Sci Rep 2019, 9, 12179). As suggested by the reviewer’s comments, I modify the sentences as follows.

Genomic alterations (EGFR mutation status (exon 18 [G719X], exon 19 [E746_A750 deletion], exon 20 [V769_V774 insertions], exon 20 [T790M], and exon 21 [L858R]) and KRAS (codon 12/13) genes) in these cell lines were previously assessed and genomic alterations were not detected in all cell lines [35].

Comment:

Line 324: Mentioning companies usually requires more information than just the name => please check the journal's guidelines in this regard. In addition, the procedure from cultivation to shipment requires further information or reference to a citable protocol to reproduce the study. EpiShear probe sonicator (cat# 53051): which company? Please indicate which experiments and/or data analyses of the study were performed by a/which commercial institution(s).

Please indicate at which steps biological and/or technical replicates were performed?

Lines 328-332: requires more detailed information about procedures (e.g. volumes, type of beads.) and buffer compositions (SDS-buffer (?)) or respective references.

Response:

I thank the reviewer for your suggestions. As suggested by the reviewer’s comment, I modified 4.2 to avoid confusion as follows.

4.2 H3K27ac CHIP-sequencing analysis

Cells were fixed with 1% formaldehyde for 15 min and quenched with 0.125 M glycine. A total of 1×107 cells were used for H3K27ac CHIP-seq. CHIP-seq analysis was performed by Active Motif Inc. (Carlsbad, CA, USA), and the detailed analysis procedures were based on the previous studies [61-63].

Briefly, chromatin was isolated after treatment with Chromatin Prep Lysis Buffer containing non-ionic detergent and protease inhibitors, followed by disruption with a Dounce homogenizer. Genomic DNA was sheared to an average length of 300-500 bp. The segments of interest were immunoprecipitated by 4 uL of specific antibody against H3K27Ac (Active Motif, cat# 39133, Lot 16119013). The protein and DNA complexes were washed, eluted from the Agarose beads and were treated with SDS buffer, RNase, and proteinase K. Crosslinks were reversed and ChIP DNA was purified by phenolchloroform extraction and ethanol precipitation.

Sequencing libraries were prepared and sequenced on Illumina’s NextSeq 500 (75 nt reads, single-end). The reads were aligned to the human genome assembly GRCh38 (hg38) by BWA (default settings) and non-duplicated mapped reads (mapping quality > 25) were used for further analysis. Alignments were extended in silico at their 3’-ends to a length of 200 bp and assigned to 32-nt bins along the genome. The histograms were stored in bigWig files and peak locations were determined using the MACS algorithm (v2.1.0) with a cutoff p-value = 1 x 10-7. ENCODE blacklist, known as false ChIP-Seq peaks, were removed. Signal maps and peak locations were used as input data to an Active Motifs proprietary analysis program.

Our data resulting from CHIP Sequencing analysis were deposited in the GEO database (accession number: GSE205455).

Comment:

Lines 372-373: how can protein expression levels of target genes be confirmed by downloading of images?

Response:

The Human Protein Atlas (https://www.proteinatlas.org) database provides detailed information on the types of antibodies used and immunohistochemical staining in each tissue. The protein expression described in the database is referred to in this paper.

Results:

Comment:

Lines 101-104: redundant with next sentence.

Response:

As suggested by the reviewer’s comment, I modified it as follows.

To investigate the dynamic epigenetic state of OSCC after long-term exposure to anticancer drugs. We identified SEs with H3K27ac peaks in control cells (HSC-3 and SAS) and cells that had been subjected to long-term cetuximab exposure (Cmab-LTE). A total of 995 and 1043 SE peaks were detected in Cmab-LTE HSC-3 and Cmab-LTE SAS cells (Figure 1).

Comment:

Lines 116-117: “Cmab-LTE cell-specific” term seems to me over interpreted => better change to “…gain peaks were identified in Cmab-LTE HSC-3 and Cmab-LTE SAS compared with the respective parental cell line.”

Response:

As suggested by the reviewer’s comment, I modified it as follows.

A total of 152 and 481 SE gain peaks were identified in Cmab-LTE HSC-3 and Cmab-LTE SAS compared with the respective parental cell lines, respectively (Figure 1 and 2).

Comment:

In my opinion, Table 1 and 2 contain too much detailed information for the main text of the manuscript. On the other hand, both contain some interesting data. I suggest merging Table 2 with Table 1 if possible, since there are some redundancies. E.g., by adding each row from Table 2 (in bold) below the corresponding rows in Table 1. Alternatively, the highly regulated genes from Table 2 could be highlighted in Table 1 (and Table 2 moved to the supplements).

Response:

I agree with reviewer’s suggestion. Moved Table 2 from the text to the Supplemental Table. Regarding the genes listed in Table 1, the genes that expression was upregulated by TCGA-OSCC tissues are shown in bold in the Table 1. I modified it as follows.

The 34 genes that expression was upregulated by TCGA-OSCC tissues are shown in bold in the Table 1, and the detail of the genes were listed in Table S1.

Comment:

Line 145: move sentence two sentences down.

Response:

As suggested by the reviewer’s comment, I modified it as follows.

Patients were divided into two groups according to the median gene expression level: high and low expression groups. The red and blue lines represent the high and low expression groups, respectively. High expression levels of all four genes significantly predicted a poorer prognosis in patients with OSCC. Nominal p-value was calculated by log-rank test.

Comment:

Section 2.4: I propose to substitute this section for section 2.3, as it represents the experimental validation of the upregulation in cell lines shown before.

Response:

I agree with reviewer’s suggestion. Swapped Section 2.3 and Section 2.4.

Comment:

Section 2.3 and 2.5: I have a feeling that this is not very robust data, as the analysis itself is based on results obtained from only two cell lines. Regarding the selection of samples/patients, the analyzed datasets are inconsistent, difficult to reconstruct, and in particular the cancer patients included in the Cancer Genome Atlas have been selected based on "convenience of collection". How many pictures were available and evaluated for this work? How were pictures selected for presentation?

Response:

There is a slight disagreement with the reviewer's comments.

To ensure the objectivity of the data, it is necessary to analyze it using large-scale and reliable data that can be accessed by anyone. The TCGA database is essential for in silico analysis, and is commonly used in cancer research articles.

Similarly, for protein expression (immunostaining), The Human Protein Atlas database ensures objectivity. The Human Protein Atlas (https://www.proteinatlas.org) database provides detailed information on the types of antibodies used and immunohistochemical staining in each tissue. The protein expression described in the database is referred to in this paper.

Comment:

Section 2.5: “C9orf89: Normal tissues displayed occasional nuclear positivity for HPA010921 whereas HPA038297 showed negative staining.” and “PISD: HPA031090 showed strong cytoplasmic immunoreactivity in both normal and cancer tissues. On the other hand, HPA031091 displayed weak positive staining in normal tissue.” => what is the result (of this study) here, except that two antibodies show different staining? How have antibodies selected for the investigation in this study?

Response:

Thank you for your suggestions. As suggested by the reviewer’s comments, I added the sentence as follows.

The moderate to high expression of each gene were confirmed on the cancer tissues, however, normal epithelial tissues were stained in case of a few antibodies.

Comment:

Line 172-174, Figure 5C: “patients with increased expression of the 4 genes” => these are roughly n=15-20 according to “mRNA High” for all 4 genes in Figure 5A but n=236 in Figure 5C. It is unclear how this group of patients is characterized.

Response:

Thank you for your suggestions. Figure 6C (the order of the figures has been changed according to reviewer’s comment) was reanalyzed and the survival rate was changed to 60 months. About the explanation of the figure. It was a misunderstanding, so I changed it as follows.

Patients with OSCC who had increased expression in at least one of the four genes showed an unfavorable survival outcome, and were characterized by aberrant cell cycle gene signatures (Figures 6C and 6D).

Comment:

Figure 6: Western blot and/or immunostaining of cell line samples would clearly substantiate the mRNA expression data.

Response:

We recognize the importance of the reviewers' suggestions, but this time we will omit this analysis. We judge that it will not have a significant impact on the outline of the article. We ask for your kind understanding and cooperation.

Discussion:

Comment:

Lines 225-231 may fit better / or be redundant with introduction – please check.

Response:

I would like to thank the reviewer for your suggestions. The first chapter has been changed as follows.

OSCC is a highly malignant cancer, and the 5-year survival of OSCC has remained below 50% [3,36]. Discovery of drug susceptibility markers and molecules involved in drug resistance is essential for improving the prognosis of patients with OSCC. A vast number of studies have shown that dysregulated epigenetic control of cancer cells is closely involved in malignant transformation, metastasis, and drug resistance [14-16].

Comment:

Lines 236-249: How does this relate to the results of this work?

Response:

I would like to thank the reviewer for your suggestions. The 2nd chapter has been changed as follows.

The super-enhancer concept is critically important in cancer research, and the identification of cancer cell-specific super-enhancers has been vigorously pursued [22,33,34,37]. A recent study using H3K27ac ChiP-seq analysis of oral cancer cell lines (HSC4 and BHY) showed that 41 genes were regulated by super-enhancers [38]. Among these genes, high expression of AHCY, KCMF1, MANBAL, and TFDP1 predicted poor prognosis of the patients with OSCC [38]. It is evident that genome-wide super-enhancer analyses provide novel information regarding OSCC/HNSCC molecular pathogenesis.

Comment:

Line 291-293: “These results suggest approaches that co-inhibit RAS-MAPK-ERK signaling and PI3K-AKT-mTOR signaling, or that combine cetuximab with immune checkpoint inhibition.” I wonder if the authors have thought about the possibility of testing this in some way in their cell line model.

Response:

I would like to thank the reviewer for your suggestions. Indeed, another study also demonstrated that co-targeting ERK and mTOR by FDA approved drugs induced a significant synergistic anti-tumor effect in OSCC preclinical models [Oncotarget. 2016; 7: 10696-10709]. We added this article for citations. As for the combination of cetuximab and immune checkpoint inhibitors, since it will take significant time to establish anexperimental system that appropriately considers the host's immune environment when using immune checkpoint inhibitors, we consider this to be a future research topic based on the findings of this study. Thus, we revised the part of Line 291-293 by adding the citation No.57 as follows.

These results suggest approaches that co-inhibit RAS-MAPK-ERK signaling and PI3K-AKT-mTOR signaling, or that combine cetuximab with immune checkpoint inhibition. Such approaches have begun to be investigated in preclinical and clinical trials, and co-inhibition of RAS-MAPK-ERK and PI3K-AKT-mTOR signaling has demonstrated antitumor effects in the HNSCC PDX model [56] and the PIK3CA+ OSCC preclinical model [57]. Clinical efficacy will be confirmed in the KURRENT study (NCT04997902).

Thank you for your constructive comments and suggestions. We believe that our manuscript has been greatly improved and is now suitable for publication in IJMS.Again, thank you for your consideration of our manuscript for publication in your journal.

Sincerely yours,

Naohiko Seki, Ph.D.

Department of Functional Genomics

Chiba University Graduate School of Medicine

1-8-1 Inohana, Chuo-ku,

Chiba 260-8670, Japan

Phone: +81-43-226-2971

Fax: +81-43-227-3442

e-mail: naoseki@faculty.chiba-u.jp

Round 2

Reviewer 1 Report

Dear Editor,

despite the fact that no clinical validation has been made so far as suggested, the text has been improved.

As a minor revision, in lines 243 and 250 genes must be written in italics.

Author Response

As suggested by the reviewer’s comment, I modified them in italic.

We would like to express our sincere gratitude for the constructive peer
reviewing of our submitted article.

Reviewer 2 Report

No further comments.

Author Response

We would like to express our sincere gratitude for the constructive peer
reviewing of our submitted article.